# Metagenomic and Antibiotic Resistance Analysis of the Gut Microbiota in *Larus relictus* and *Anatidae* Species Inhabiting the Honghaizi Wetland of Ordos, Inner Mongolia, from 2021 to 2023

**DOI:** 10.3390/microorganisms12050978

**Published:** 2024-05-13

**Authors:** Ronglei Huang, Xue Ji, Lingwei Zhu, Chengyang Zhang, Tingting Luo, Bing Liang, Bowen Jiang, Ang Zhou, Chongtao Du, Yang Sun

**Affiliations:** 1State Key Laboratory for Diagnosis and Treatment of Severe Zoonotic Infectious Diseases, Key Laboratory for Zoonosis Research of the Ministry of Education, Institute of Zoonosis, College of Veterinary Medicine, Jilin University, Changchun 130062, China; hrl17835714615@163.com (R.H.); zcy1123101929@163.com (C.Z.); 13290348002@163.com (A.Z.); 2Changchun Veterinary Research Institute, Chinese Academy of Agricultural Sciences, Changchun 130122, China; ji_xuecn@aliyun.com (X.J.); lingweiz@126.com (L.Z.); liangbing0427@yeah.net (B.L.); jiangbowen17@126.com (B.J.); 3Key Laboratory of Jilin Province for Zoonosis Prevention and Control, Changchun 130122, China; 4College of Animal Sciences, Jilin University, Changchun 130062, China; 15590676136@163.com

**Keywords:** *Anatidae*, biofilm, metagenomic, *Escherichia coli*, gut microbiota, *Larus relictus*

## Abstract

Gut microbes thrive by utilising host energy and, in return, provide valuable benefits, akin to a symbiotic relationship. Here, metagenomic sequencing was performed to characterise and compare the community composition, diversity and antibiotic resistance of the gut microbiota of Relict gull (*Larus relictus*) and *Anatidae* species. Alpha diversity analysis revealed that the intestinal microbial richness of *L. relictus* was significantly lower than that of *Anatidae*, with distinct differences observed in microbial composition. Notably, the intestines of *L. relictus* harboured more pathogenic bacteria such as clostridium, which may contribute to the decline in their population and endangered status. A total of 117 strains of *Escherichia coli* were isolated, with 90.60% exhibiting full susceptibility to 21 antibiotics, while 25.3% exhibited significant biofilm formation. Comprehensive Antibiotic Resistance Database data indicated that glycopeptide resistance genes were the most prevalent type carried by migratory birds, alongside quinolone, tetracycline and lincosamide resistance genes. The abundance of resistance genes carried by migratory birds decreased over time. This metagenomic analysis provides valuable insights into the intestinal microbial composition of these wild bird species, offering important guidance for their conservation efforts, particularly for *L. relictus*, and contributing to our understanding of pathogen spread and antibiotic-resistant bacteria.

## 1. Introduction

Birds are one of the most diverse groups of vertebrates globally, with >10,000 species distributed across almost every corner of the world [1]. Due to their extensive flight distances, wide-ranging movements, and intricate migration routes, wild birds are considered sentinels for monitoring the dissemination of antibiotic-resistant bacteria. They can acquire antibiotic-resistant bacteria or zoonotic pathogens through consuming untreated sewage or scavenging on unprocessed waste materials. Consequently, they may transmit these microorganisms to humans or contaminate the environment along their migratory paths [2]. Therefore, investigating the composition of gut microbiota in wild birds holds significant potential for impeding the spread of associated pathogenic microorganisms and plays a crucial role in conserving and managing wild avian populations.

The gut microbes of many wild birds have been studied, including Sandhill cranes (*Grus canadensis*) [3] and Whooper swans (*Cygnus cygnus*) [4]. These significant research findings provide valuable insights for the development of policies aimed at safeguarding endangered species. Relict gull (*Larus relictus*) is an indigenous species commonly found in Inner Mongolia that has been classified as a first-class protected animal in China and is considered vulnerable by the International Union for the Conservation of Nature [5]. This species primarily inhabits saltwater bodies and alkaline lakes situated between 1200 and 1500 m above sea level within elevated regions. Thriving under desert–semi-desert ecological conditions makes them unique among avian species. There exist four relatively independent breeding populations worldwide, of which Erdos City in Inner Mongolia is the largest and most important for maintaining global numbers [6]. *Anatidae* collectively refer to *Anseries* birds belonging to the order *Anseriformes*, a diverse avian family constituting ducks, geese and swans [7], with a wide range of species and abundance in Inner Mongolia, and interesting group dynamics and distribution patterns.

*Anatidae* and *L. relictus* coexist within the Ordos Honghaizi Wetland, with the latter a crucial protected species, while *Anatidae* are the largest breeding population of migratory birds in this region. Studies show that both environment and dietary nutrition influence animal gut microbial communities [8,9,10,11]. Despite sharing similar dietary habits and habitats within the Ordos Honghaizi Wetland, *Anatidae* and *L. relictus* are genetically distinct. This study aims to provide valuable insights for conservation efforts concerning these important avian species, focusing on *L. relictus*, which holds first-class protected status, while also offering potential insights into transmission pathways of associated pathogenic microorganisms.

*Escherichia coli* serves as an excellent bacterial model for investigating the dissemination of antibiotic resistance. While most *E. coli* strains are part of the normal intestinal flora in migratory birds, they can induce disease under specific circumstances and are collectively referred to as avian pathogenic *E. coli*. One study demonstrated the pivotal role of horizontal gene transfer in conferring antibiotic resistance within biofilms [12]. Horizontal gene transfer facilitates the exchange of antibiotic resistance genes (ARGs) among bacterial species and strains through mobile genetic elements such as plasmids, transposons and integrons. This mechanism promotes the dissemination of resistance genes within biofilm-associated bacterial populations, ultimately leading to the emergence of multidrug-resistant strains. Furthermore, biofilm formation enhances the acquisition of antibiotic-resistant genes, thereby impeding effective treatment with antibiotics [13].

Characterisation of the gut microbial community in migratory birds has contributed to our understanding of their microbial origins and the risk of pathogenic bacteria spreading to the environment. Identifying pathogenic bacteria with multiple hosts could be used to evaluate the potential for dissemination from migratory birds to other species. However, research on the intestinal flora of *L. relictus* and *Anatidae* is limited, as are data for systemic antibiotic resistance analysis. In this study, metagenomic sequencing was conducted on faecal samples from *L. relictus* and *Anatidae* to determine species composition and differences in the gut microbial community. Additionally, systemic antibiotic resistance analysis was performed. The findings are of significance for the protection of precious migratory bird species and provide a theoretical basis for disease prevention and control of wildlife.

## 2. Materials and Methods

### 2.1. Study Area and Sample Collection

The Ordos Honghaizi Wetland is located ~16 km south of Ordos City in the Inner Mongolia Autonomous Region and is the main breeding ground of the *L. relictus* population. The Ordos Honghaizi Wetland is located ~16 km south of Ordos City in the Inner Mongolia Autonomous Region and is the main breeding ground of the *L. relictus* population. The water area of the Ordos Honghaizi Wetland has steadily expanded year over year, while the water quality has gradually improved. Consequently, it serves as a crucial component in the construction of the ecological security system in the central urban area. From 2021 to 2023, 369 samples of *L. relictus* faeces, 361 samples of *Anatidae* faeces and 15 environmental samples were collected in the areas (109°7′99″ E to 109°8′35″ E; 39°5′76″ N to 39°6′43″ N). The faecal samples were collected in April 2021 and 2023 during the spring season, and in October 2022 during the autumn season. In order to be less exposed to the effects of faeces, we took fresh upper-layer faeces samples with a disposable sterile cotton swab, placed them in a 2 mL sterile centrifuge tube, transported them to the laboratory with an ice pack and stored them at −80 °C for further experiments.

### 2.2. DNA Extraction and Metagenomic Analysis

Genomic DNA was randomly sheared into short fragments, and sequencing libraries were generated. The purity and integrity of DNA were assessed using 2% agarose gel electrophoresis. Subsequently, qualified DNA samples were randomly fragmented using a Covaris ultrasonic crusher (Covaris, LLC., Woburn, MA, USA) with a target size of 350 bp. The obtained fragments were end-repaired, A-tailed and further ligated with Illumina adapter. The fragments with adapters were PCR-amplified, size-selected and purified. The library was checked using Qubit and real-time PCR for quantification and using an Agilent 2100 bioanalyzer (Agilent, Santa Clara, CA, USA) for size distribution assessment. Quantified libraries were pooled and sequenced on Illumina platforms, according to the effective library concentration and data amount required.

Alpha diversity analysis was conducted to evaluate the richness and diversity of bacterial species. Bray–Curtis distance-based scatter plots were generated using NMDS and Cluster Tree methods to identify differences in bacterial composition between *L. relictus* and *Anatidae*. Linear discriminant analysis effect size (LEfSe) was employed to determine significant differences in intestinal microbiota composition among surviving *L. relictus* and *Anatidae*. The number and types of antibiotic resistance genes carried by surviving *L. relictus* and *Anatidae*, as well as in environment samples, were analysed using the Comprehensive Antibiotic Resistance Database (CARD; https://card.mcmaster.ca/, accessed on 27 July 2023).

### 2.3. Data Analysis

DIAMOND software (https://github.com/bbuchfink/diamond/, accessed on 27 July 2023) [14] was used for alignment of unigene sequences with those of bacteria, fungi, archaea and viruses extracted from the NCBI NR database (https://www.ncbi.nlm.nih.gov/, accessed on 27 July 2023). Unigenes were aligned to the CARD database (https://card.mcmaster.ca/, accessed on 27 July 2023) [15] using Resistance Gene Identifier (RGI) software (version 6.0.2) [16] provided by the CARD database [17]. According to the RGI alignment results and unigene abundance information, the relative abundance of each Antibiotic Resistance Ontology (ARO) was calculated. Based on ARO abundance, abundance histograms, abundance clustering heatmaps, abundance distribution Chord diagram, ARO difference analysis between groups, resistance genes (unigenes annotated as ARO) and species attribution analysis of resistance mechanisms were carried out. The prediction of mobile genetic elements involves utilising Blast software (version 2.15.0) to compare unigene sequences with each transfer element database, namely INTEGRALL (Integrall: http://integrall.bio.ua.pt/, accessed on 25 January 2024), Plasmid (https://ccb-microbe.cs.uni-saarland.de/plsdb/, accessed on 25 January 2024) and ISfinder (https://isfinder.biotoul.fr/about.php, accessed on 25 January 2024). The comparison parameter for identity value was set at 95%, while the e-value was set at 1 × 10^−5^. Subsequently, the abundance of unigenes was mapped to annotated transfer elements and utilised as their respective abundances for visual analysis. The functions of significantly abundant bacteria were predicted using the KEGG database.

### 2.4. Isolation and Identification of E. coli

Each faecal sample weighing ~1 g was placed in a 1.5 mL centrifuge tube and resuspended in 1 mL sterile saline solution (Liaoning Minkang Pharmaceutical Co., Ltd., Dalian, China) via thorough agitation and mixing. Subsequently, 600 μL of the faecal supernatant was collected and combined with 400 μL of a 50% glycerol (Guangdong Guanghua Sci-Tech Co., Ltd., Shantou, China) saline solution for preservation, before being stored at −80 °C.

For turbidity treatment of water samples, 20 mL of relatively clear water was passed through a detachable filter membrane with a pore size of 0.22 μm. The filter membrane was then transferred to a 4 mL centrifuge tube and washed with 2 mL sterile saline solution. Following this step, we retained and mixed together 600 μL of the resulting supernatant with another aliquot consisting of 400 μL of preservation solution containing both glycerol (50%) and saline, prior to storing it at −80 °C.

Soil samples weighing 2–3 g were individually placed in separate tubes measuring up to 4 mL. To ensure proper agitation and mixing, 2 mL of sterile saline solution was added to each tube. A 600 μL aliquot of the resulting supernatant was combined with 400 μL of sample derived from preservation solution composed of 50% glycerol and 50% saline, and mixtures were stored at −80 °C.

For each sample, 100 μL of storage solution was added to 900 μL of Gram-negative enrichment solution liquid medium (Qingdao Hi-Tech Industrial Park Hope Bio-Technology Co., Ltd., Qingdao, China) and incubated at 37 °C for 12 h. A 20 μL volume of enrichment solution was spotted onto a Macconkey agar plate (Qingdao Hi-Tech Industrial Park Hope Bio-Technology Co., Ltd.) and incubated at 37 °C for 16–18 h, resulting in large, round, peachy, opaque colonies. Single colonies of suspected *E. coli* were purified until they were free of miscellaneous bacteria, and only one strain was isolated from each sample.

*E. coli*-specific 16S rDNA was synthesised using primers (Appendix A) synthesised by Jilin Kume Biotechnology Co., Ltd. (Changchun, China). Positive (ATCC 25922) and negative (deionised water) controls were included. PCR amplification was performed using Taq DNA Polymerase (TaKaRa, Tokyo, Japan) with an initial denaturation step at 94 °C for 5 min for cell lysis, followed by 30 cycles at 94 °C for 40 s, 60 °C for 40 s and 72 °C for 1 min, then a final extension step at 72 °C for 7 min. PCR products were analysed by electrophoresis on a 1% (*w*/*v*) agarose gel.

### 2.5. Antimicrobial Susceptibility Testing

Isolates were subjected to species identification and antibiotic susceptibility testing using a BD Phoenix-100 Automated Microbiology System (Becton, Dickinson and Co., Franklin Lakes, NJ, USA). A total of 21 antibiotics from 13 classes were tested, namely amikacin (AMK), gentamicin (GEN), imipenem (IPM), meropenem (MEM), cefazolin (CFZ), ceftazidime (CAZ), cefotaxime (CTX), cefepime (FEP), aminotrenam (AZM), ampicillin (AMP), piperacillin (PIP), amoxicillin/clavulanic acid (AMC), ampicillin/sulbactam (SAM), piperacillin/tazobactam (TZP), colistin (CST), methotrexate/sulphamethoxazole (SXT), chloramphenicol (CHL), ciprofloxacin (CIP), levofloxacin (LVX), moxifloxacin (MXF) and tetracycline (TET). Resistance phenotypes were determined according to the Clinical and Laboratory Standards Institute, and identification results showing S, I and R indicated that this strain exhibited susceptibility, intermediate and resistance phenotypes to the antimicrobial drug [18]. Strains that acquired resistance to at least one drug from each of the three or more classes of antimicrobial drugs based on the resistance phenotype results were considered to be multidrug-resistant bacteria.

### 2.6. Biofilm Assay

Biofilm formation by the *E. coli* isolates was assessed using the microtiter plate method as described previously [19] with some modifications. The turbidity of the bacterial solution of overnight-cultured *E. coli* was adjusted to 0.5 McFarland Standard, and 200 μL of bacterial solution was taken after 100-fold dilution and added to 96-well cell culture plates (Costar, Washington, DC, USA). The plates were incubated for 24 h at 37 °C. Positive control wells contained *E. coli* ATCC 25922, and negative control wells contained uninoculated Luria–Bertani medium (Qingdao Hi-Tech Industrial Park Hope Bio-Technology Co., Ltd.). Studies were performed in triplicate for each well. The contents of each well were discarded, wells were washed three times with sterile phosphate-buffered saline (PBS; Beijing Solarbio Science & Technology Co., Ltd., Beijing, China), air-dried at room temperature and stained with 200 μL of 1% Crystal Violet for 5 min. Wells were then washed three times with PBS to remove excess stain and dried at room temperature. Dye bound to adherent cells was resolubilised in 200 μL of 95% ethanol. The optical density (OD) value of the biofilm was determined at 570 nm using an Enzyme standard instrument (BioTek Instruments Inc., Winooski, VT, USA), and the average OD of each duplicate result was taken. All strains were classified into four categories according to the normalised OD: nonadherent (OD ≤ OD_C_); weakly adherent (OD_C_ < OD ≤ 2 × OD_C_); moderately adherent (2 × OD_C_ < OD ≤ 4 × OD_C_); strongly adherent (4 × OD_C_ < OD). Optical density cut-off value (ODc) = average OD of negative control + 3 × standard deviation (SD) of negative control [20].

## 3. Results

### 3.1. Alpha and Beta Diversity Analyses

Analysis of alpha diversity revealed significant differences in Chao1 and ACE indices (*p* < 0.05) between *L. relictus* and *Anatidae*, indicating a distinct variation in gut bacteria diversity (Figure 1A,B). These findings highlight the substantial dissimilarity in microbial composition between *L. relictus* and *Anatidae*.

Beta diversity analysis was performed to assess variations in bacterial composition between *L. relictus* and *Anatidae*. Nonmetric multidimensional scaling (NMDS) scatter plots and cluster trees based on Bray–Curtis distances were generated for visualisation purposes. Notably, faecal samples from different individuals within the same bird species exhibited clear clustering patterns, further supporting the comparative data that demonstrated a marked difference in microbial composition between *L. relictus* and *Anatidae* (Figure 2A,C). Additionally, distinct clustering trends were observed among samples collected across different years (Figure 2C). Significance testing using analysis of similarities (ANOSIM) confirmed a significant disparity in gut bacteria composition between *L. relictus* and *Anatidae* (Figure 2B).

### 3.2. Composition of the Gut Microbiota Community in L. relictus and Anatidae

The bacterial composition of each sample was examined, and the top 10 phyla/genera were used to plot histograms. The dominant phyla in *Larus relictus* were Firmicutes, Proteobacteria and Actinobacteriota, compared with Proteobacteria, Firmicutes, Bacteroidetes and Actinobacteriota in *Anatidae*. The most dominant phylum in *L. relictus* was Firmicutes, compared with Proteobacteria in *Anatidae* (Figure 3A,B). The most dominant genus in *L. relictus* was *Ligilactobacillus*, compared with *Psychrobacter* in *Anatidae* (Figure 3C,D). It is noteworthy that the percentage of *Clostridium* in *L. relictus* was higher than in *Anatidae*. Among them, *C. perfringens* causes necrotic enteritis in poultry, which has a serious impact on the poultry industry [21]. This may be an important pathogen affecting the survival of *L. relictus*.

The gut microbiota composition of *L. relictus* and *Anatidae* was analysed to identify significant differences between the two groups. Metastat analysis results revealed that, at the phylum level (Figure 4A), there was a significantly higher abundance of Bacteroidetes, Cyanobacteriota, Verrucomicrobia, Thermodesulfobacteriota, Ignavibacteriota and Fibrobacterota in *Anatidae* than in *L. relictus*. At the genus level (Figure 4B), there was a substantially higher abundance of *Flavobacteriaceae*, *Cereibacter*, *Helicobacter*, *Polaromonas*, *Comamonas* and *Bacteroides* in *Anatidae* than in *L. relictus* (*p* < 0.01).

### 3.3. Application of ARGs in Intestinal Microbes of Migratory Birds

Using data from the Comprehensive Antibiotic Resistance Database, we identified 252 resistance genes across 46 sets of stool samples (Figure 5A). The migratory bird samples were categorised based on their collection time. Among these samples, 76 resistance genes were consistent with those detected in the environment, with glycopeptide resistance genes being the most prevalent. This finding suggests that the higher prevalence of glycopeptide resistance genes among migratory bird samples may be attributed to their exposure to environmental sources harbouring such genetic elements. Furthermore, our analysis unveiled a temporal decline in the prevalence of resistant gene carriage among migratory birds, which may be linked to the implementation of a comprehensive ban policy enforced in 2020 (Figure 5B).

Heatmaps were generated to visualise the distribution of the top 30 ARGs in each sample (Figure 5C). Subsequently, cluster analysis was conducted to examine the ARG profiles carried by *L. relictus* and *Anatidae*, followed by the generation of heatmaps (Figure 5D). The findings revealed a relatively low prevalence of antibiotic resistance genes among *L. relictus* and *Anatidae*, as well as their respective habitats, which could potentially be attributed to the comprehensive prohibition policy implemented in China in 2020. Furthermore, *Anatidae* harboured more resistance genes than *L. relictus*. Notably, glycopeptide resistance genes were predominantly found in migratory birds.

To visually depict the abundance ratio of ARGs in all samples, a selection was made to draw a Chord diagram featuring the top 30 ARGs (Figure 5E). The figure illustrates that antibiotic resistance genes carried in all samples predominantly belong to glycopeptides, ammonium salts, macrolides and lincomycins. A Chord diagram showing the distribution of resistance mechanisms based on the mechanism of action of resistance genes relative to species is shown in Figure 5F. It is evident that *Pseudomonas* (Proteobacteria) and *Bacillus* (Firmicutes) primarily exhibit antibiotic action, antibiotic inactivation, and antibiotic target change as their main resistance mechanisms.

### 3.4. Prediction of Mobile Genetic Elements

It is crucial to note that according to the ISfinder database, the insertion sequences (ISs) exhibited greater abundance in faecal samples from *L. relictus* and *Anatidae* than in environmental samples (Figure 6A). According to the INTEGRALL database, our analysis detected integrases and gene cassettes predominantly in environmental samples. Meanwhile, other mobile genetic elements were abundant in *L. relictus*, and *Anatidae* faecal and environmental samples were mainly ‘integral’ (Figure 6B) and ‘plasmid’ types (Figure 6C).

### 3.5. Isolation and Identification of E. coli and Analysis of Antibiotic-Resistant Phenotypes

A total of 117 *E. coli* strains were isolated from 561 migratory bird faecal samples from 2021 to 2023. The isolates were characterised in terms of antibiotic sensitivity using an automated microbiology system. The resistance patterns are presented in Appendix A. The isolation rate of multi-resistant *E. coli* was 5.98% (7/117), the isolation rate of *E. coli* strains that were fully susceptible to the 21 antibiotics tested was 90.60% (106/117), the most tetracycline-resistant strain was found in the 117 *E. coli* strains, and the isolation rate of *E. coli* with a tetracycline-resistant phenotype was 6.83% (8/117).

### 3.6. Biofilm Formation by E. coli

In this study, *E. coli* isolates were categorised into four groups according to the biofilm-forming strength (Appendix A). Weak biofilm producers accounted for 18.77% (22/117) of the isolates; 5.98% (7/117) demonstrated moderate capabilities for biofilm production; 0.85% (1/117) demonstrated strong capabilities for biofilm production; 74.7% (87/117) did not form significant biofilm.

### 3.7. Bacterial Community Functional Prediction

The functions of bacteria with significant abundance in *L. relictus, Anatidae* and environmental samples were predicted using the Kyoto Encyclopedia of Genes and Genomes (KEGG) database. At the Class 1 level, functional prediction revealed that the bacterial community exhibited a predominant abundance of genes associated with cellular processes, environmental information processing, human diseases, metabolism and organismal systems (Figure 7A). The distribution of KEGG subclasses (level 2) indicated a higher occurrence of genes associated with various human diseases, followed by organismal systems, circulatory systems and metabolism (Figure 7B). The distribution of KEGG subclasses (level 3) indicated a higher occurrence of genes associated with various aspects of environmental information processing, followed by membrane transport, ABC transporters, signal transduction and two-component systems. These functions were significantly elevated in *Anatidae* compared with *L. relictus* (Figure 7C).

## 4. Discussion

Birds can acquire antibiotic-resistant bacteria or zoonotic pathogens through the ingestion of untreated sewage and garbage, which can subsequently be transmitted to humans or contaminate the environment along their flight paths [2]. Therefore, research on avian gut microbiota has garnered significant attention as it offers valuable insights for safeguarding endangered bird species. When studying microbial transmission, two intriguing host types are maintenance hosts and bridge hosts [22]. Maintenance host microbes persist within the host even without transmission from other hosts, whereas bridging hosts do not maintain microbes for extended periods but can disseminate them from maintenance hosts to other populations. Migratory birds serve as both types of hosts when spreading antibiotic-resistant bacteria, playing a pivotal role in the horizontal dissemination of associated antibiotic-resistant strains. The resistance phenotype of *E. coli*, a well-established bacterial model for investigating the dissemination of antibiotic resistance, was assessed in this study. In general, both *L. relictus* and *Anatidae* exhibited a lower prevalence of resistance genes, with only 5.98% (7/117) of isolated *E. coli* strains showing multi-resistance. Furthermore, these quinolone- and tetracycline-resistant *E. coli* strains were identified in the metagenomic resistance data along with their corresponding counterparts such as *qacG*, *qacJ* and *tetW.* These findings were further supported by the presence of corresponding resistance genes in the macrogenomic data. Consequently, investigating bird gut microbiota can illuminate the composition of the gut flora and resistance genes, potentially impeding the propagation of pathogenic microorganisms and antibiotic-resistant bacteria associated with birds. This study establishes a theoretical foundation for managing and conserving avian populations.

In this study, metagenomic sequencing was performed on 46 faecal samples obtained from *L. relictus* and *Anatidae*. Alpha diversity analysis revealed significant disparities in Chao1 and ACE indices (*p* < 0.01) between these two groups. For beta diversity analysis, scatter plots were generated using NMDS and Cluster Tree based on the Bray–Curtis distance metric. Distinct clustering patterns were observed among different faecal samples within each bird species, as well as across different years, indicating that population dynamics, temporal variations and environmental factors exert influence over gut microbial composition. At the phylum level, the dominant phyla in the *L. relictus* population were Firmicutes, Proteobacteria and Actinobacteriota, consistent with previous findings [23]. In the *Anatidae* group, Proteobacteria, Firmicutes, Bacteroidetes and Actinobacteriota were the predominant phyla. The low-temperature bacteria identified included *Psychrobacter*, *Flavobacterium* and *Pseudomonas* [24,25]. This could be attributed to faecal samples being collected during spring and fall seasons, when temperatures are generally low in the Inner Mongolia region. Hence, it is plausible that migratory bird gut microorganisms are dominated by low-temperature bacteria during these two seasons. Additionally, lactic acid bacteria play a crucial role in regulating the gut health of birds, as observed in this and previous studies [26]. Notably, a high abundance of pathogenic bacteria such as *Clostridium* within the *L. relictus* intestinal tract may contribute to its declining population, leading it towards endangered status [27,28].

Analysis using the Comprehensive Antibiotic Resistance Database revealed relatively few antibiotic resistance genes carried by both *L. relictus* and *Anatidae* along with their habitats, possibly due to China’s implementation of a total antibiotic drug ban policy since 2020. *Anatidae* harboured more resistance genes than *L. relictus*, with glycopeptide resistance genes most predominant among migratory bird samples, including 76 resistance genes shared with their environment where glycopeptide resistance genes were most prevalent. Migratory birds generally do not come into direct contact with human beings. The isolation rate of multi-resistant *E. coli* in this study was 5.98% (7/117), which indicates very few *E. coli* carrying antibiotic-resistant phenotypes in migratory birds. These multi-resistant strains may be acquired by migratory birds from the ecosystem, which is a side effect of the good environmental protection of Erdos, Inner Mongolia, which avoids excessive antibiotic release into the environment. The extracellular polymeric substances secreted by biofilm-forming bacteria may protect these bacteria from antimicrobial agents in various ways, such as chemically reacting with and neutralising antimicrobial agents, or by creating a diffusion barrier [29]. For example, a *Staphylococcus aureus* strain harbouring a plasmid encoding gentamicin resistance was also reportedly resistant to the cationic disinfectant propamidine [30]. In this study, 74.4% of *E. coli* isolated were incapable of forming biofilms, which reflects that biofilms and antibiotic resistance are not yet a serious problem in migratory birds. The role of horizontal gene transfer in biofilm-associated antimicrobial resistance has recently been reported [13]. Mobile elements such as plasmids, transposons and integrons can transfer antibiotic resistance genes between bacterial species and strains via horizontal gene transfer. Therefore, there is a need to understand the mechanisms of antimicrobial resistance associated with biofilms.

However, metagenomic sequencing also has some disadvantages. It is relatively expensive, which often necessitates the pooling of collected samples for cost considerations, making it difficult to obtain specific data for a single sample due to mixed processing. Moreover, when dealing with samples with a high host background, challenges may arise in effectively removing host cells or nucleic acids using metagenomic sequencing techniques due to a lack of genome sequence data for wildlife. Currently, metagenomic binning analysis enables the assembly of individual bacterial genomes and facilitates gene and functional annotation as well as comparative genomic and evolutionary analyses at the strain level. This capability holds great significance for identifying novel pathogens of wildlife. Despite its limitations, the potential and value of metagenomic research will continue to be explored and harnessed through technological advancements and expanded application domains.

In summary, we conducted metagenomics sequencing to analyse faecal samples of *L. relictus* and *Anatidae*. Our findings revealed that although there were differences in the intestinal flora composition between these bird species, they were predominantly dominated by Firmicutes, Proteobacteria and Actinobacteriota. These differences may be attributed to seasonal variations as well as intergroup disparities. However, this is not an absolute explanation; for example, Song et al. [31] showed that in most non-flying mammals, there is a strong correlation between microbial community similarity, host diet and host phylogenetic distance. In birds, however, the correlation between gut microbiome and diet or host phylogeny is very weak, and many microorganisms are shared across species with little correlation to diet or host affinity. Notably, distinct clustering patterns were observed among faecal samples from birds of the same species and also based on samples collected in different years. Compared with *Anatidae*, *L. relictus* harboured a higher abundance of clostridia, potentially contributing to the decline in populations of this species. The prevalence of antibiotic-resistant genes carried by migratory birds decreased over successive years. Comparative analysis of the gut microbiota of different bird species in this study could help to reveal the factors affecting their microbial composition and provide a theoretical basis for avian conservation efforts including disease prevention and control.

## Figures and Tables

**Figure 1 microorganisms-12-00978-f001:**
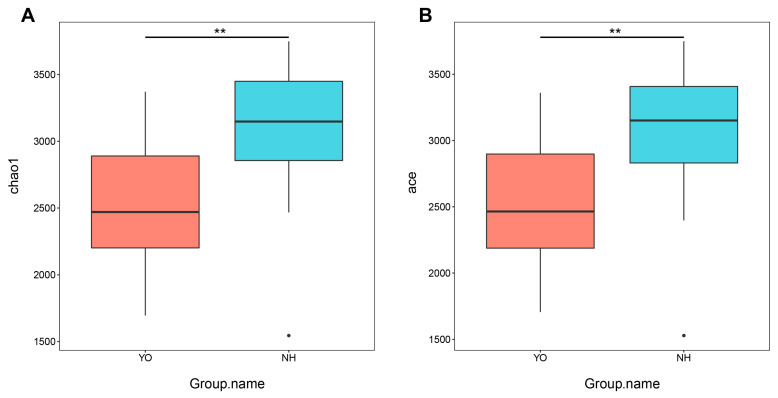
Gut bacterial alpha diversity. (**A**) Chao1 diversity. (**B**) ACE index of bacteria genera in each sample. YO: *L*. *relictus*; NH: *Anatidae*. ** *p*  < 0.01.

**Figure 2 microorganisms-12-00978-f002:**
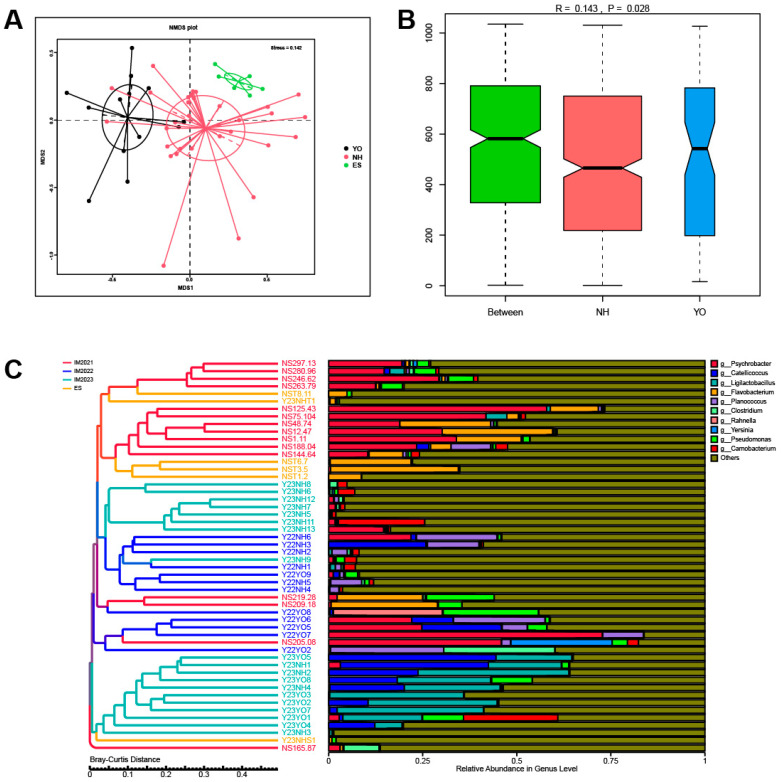
Gut bacterial beta diversity. (**A**) Bray–Curtis distance nonmetric multidimensional scaling plots. (**B**) Analysis of similarities (ANOSIM) of bacterial genera in each sample. (**C**) Bray–Curtis Distance Cluster Tree Analysis of relative abundance at the genus level in each sample. YO: *L. relictus*; NH: *Anatidae*; ES: environmental samples.

**Figure 3 microorganisms-12-00978-f003:**
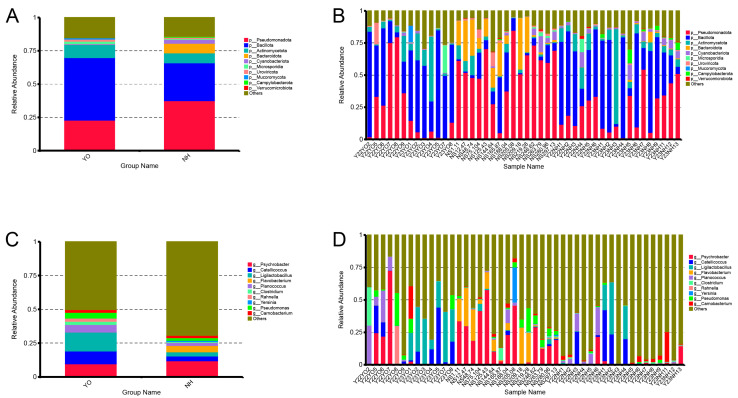
Bar graph of the relative bacterial abundance at phylum and genus levels. The bacterial composition of each sample was examined, and the top 10 phyla/genera were used to plot histograms. (**A**,**C**) Bar graphs of the relative bacterial abundance at the phylum level. (**B**,**D**) Bar graphs of the relative bacterial abundance at the genus level. YO: *L*. *relictus*; NH: *Anatidae*.

**Figure 4 microorganisms-12-00978-f004:**
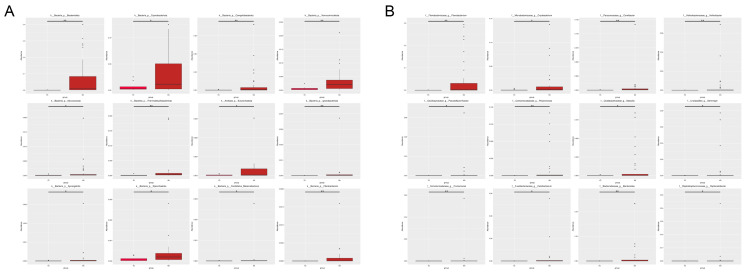
Metastat analysis of species’ differences between groups. (**A**) Metastat analysis of species’ differences at the phylum level. (**B**) Metastat analysis of species’ differences at the genus level. YO: *L. relictus*; NH: *Anatidae*. * *p*  < 0.05; ** *p*  < 0.01.

**Figure 5 microorganisms-12-00978-f005:**
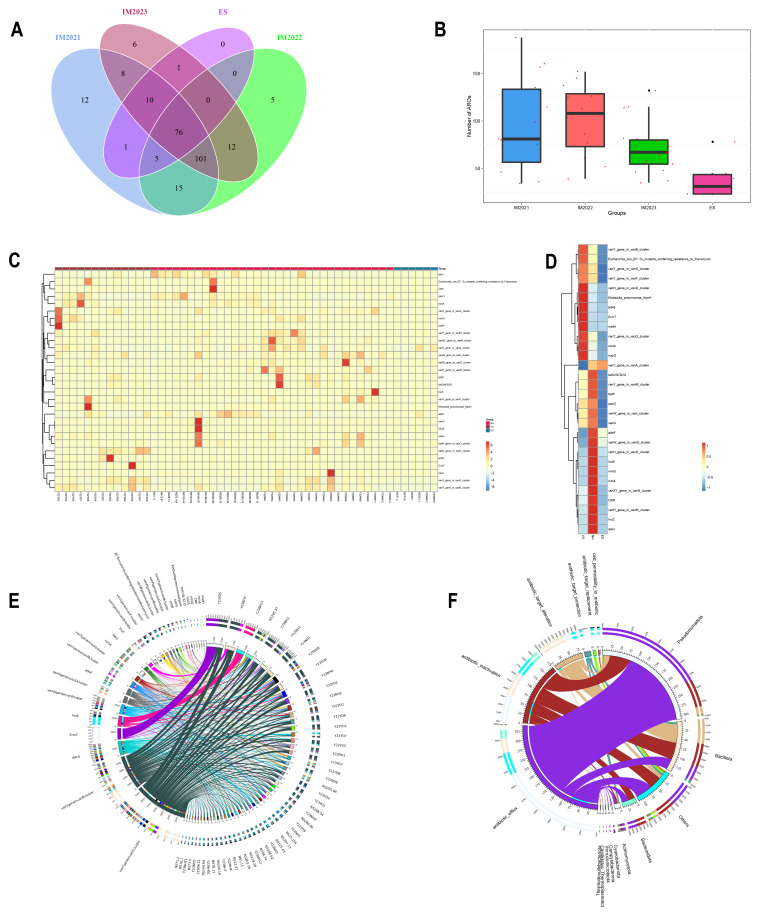
Application of antibiotic resistance genes in intestinal microbes of migratory birds. (**A**) Wayne plots of three groups of migratory bird faecal samples versus environmental samples carrying resistance genes from 2021 to 2023, grouped by time. (**B**) Box plots of differences in carrying resistance genes between three groups of migratory bird faecal samples and environmental samples from 2021 to 2023, using time as a subgroup. (**C**) Clustering heatmap of the top 30 resistance genes for each sample. (**D**) Clustering heatmap for *L. relictus*, *Anatidae* and environmental carriage of the top 30 resistance genes. (**E**) Plotting Chord diagram illustrating the antibiotic resistance ontologies with maximum abundance of the top 30 resistance genes. (**F**) Chord diagram showing the distribution of resistance mechanisms based on the mechanism of action of resistance genes relative to species. It is worth noting that the outermost ring features the nomenclature of resistance genes, with each gene distinctly labelled by a unique colour. The length of the band on this ring represents the relative abundance or expression level of the resistance genes across various samples or groups. The interconnecting lines bridging the layers of rings serve to delineate the association relationships among the resistance genes. The thickness and colouration of these lines provide insights into the strength and nature of these associations, respectively, enabling a deeper understanding of the intricate interplay among the genes. YO: *L. relictus*; NH: *Anatidae*. ES: environmental samples.

**Figure 6 microorganisms-12-00978-f006:**
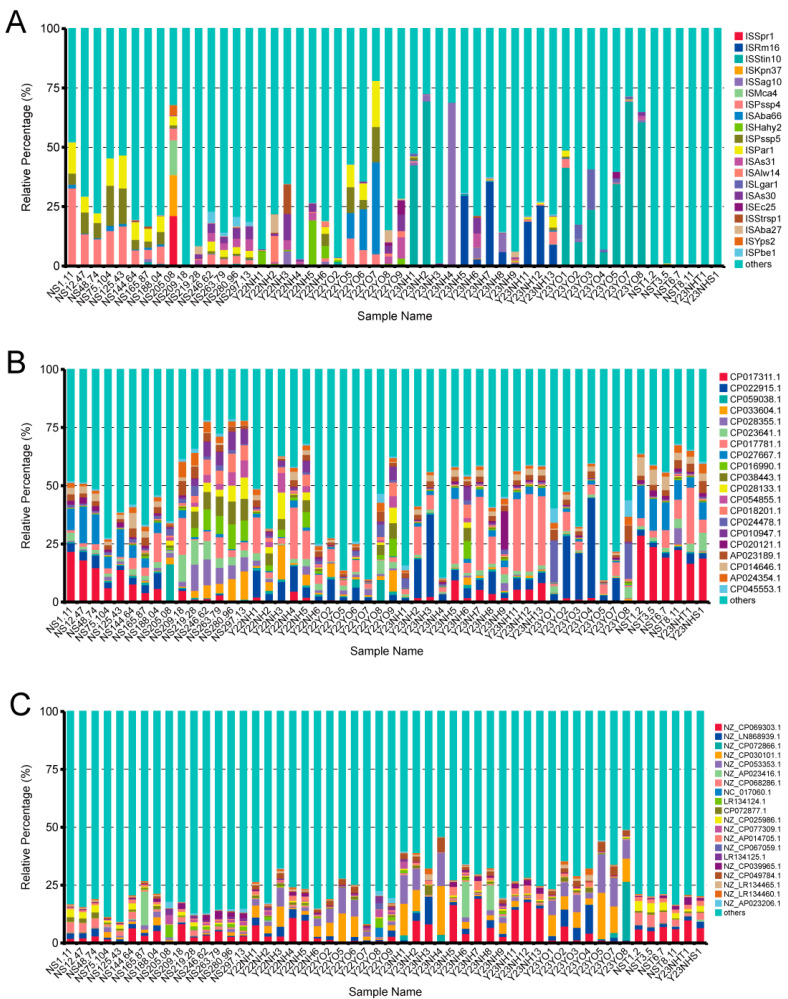
Mobile genetic element prediction for *L. relictus* and *Anatidae* (faecal) and environmental samples. (**A**) Relative abundance of the top 20 Insertion Sequences (ISs) among samples. (**B**) Relative abundance of the top 20 integrons (integrases and gene cassettes) across samples. (**C**) Relative abundance of the top 20 plasmids among samples.

**Figure 7 microorganisms-12-00978-f007:**
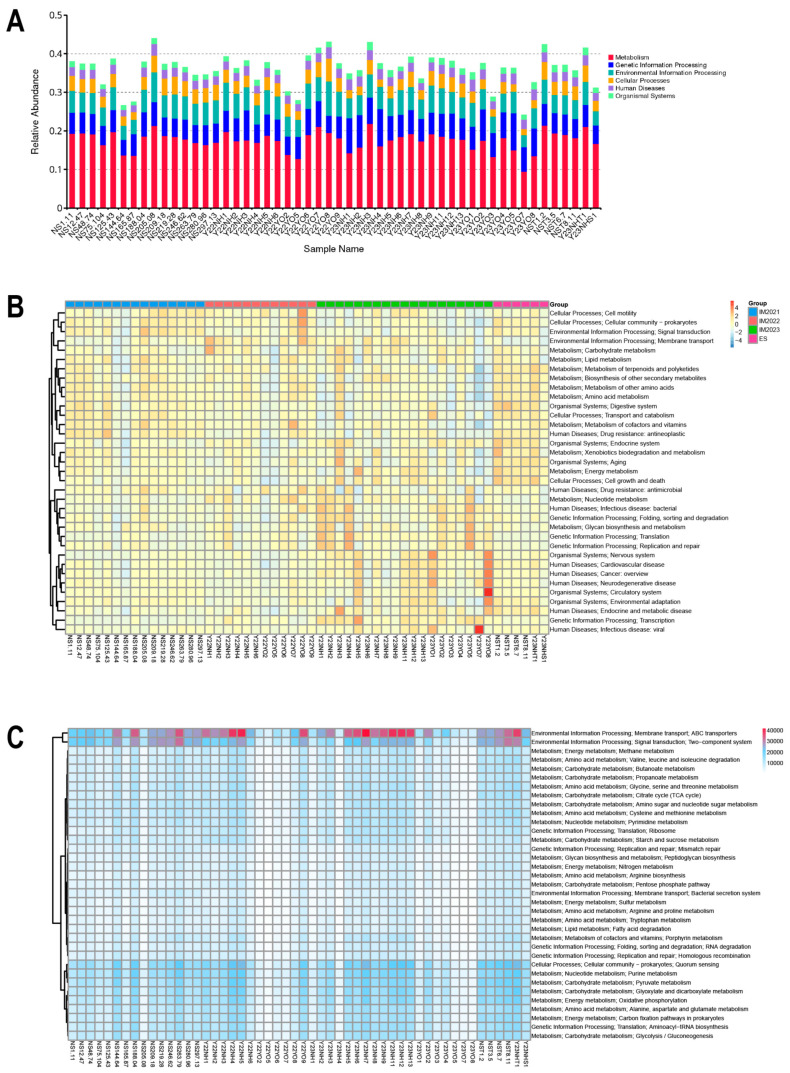
Bacterial community functional prediction using the Kyoto Encyclopedia of Genes and Genomes (KEGG) database. (**A**) Class 1 level. (**B**) Class 2 level. (**C**) Class 3 level.

## Data Availability

All data have been deposited in the NCBI repository under accession number PRJNA1063657.

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
