# Peer review of "Metagenomic and Antibiotic Resistance Analysis of the Gut Microbiota in Larus relictus and Anatidae Species Inhabiting the Honghaizi Wetland of Ordos, Inner Mongolia, from 2021 to 2023"

_microorganisms, 2024, doi:10.3390/microorganisms12050978_

Round 1
Reviewer 1 Report
Comments and Suggestions for Authors
The manuscript is a comprehensive work investigating the gut microbiota composition and antibiotic resistance profiles of Larus relictus and Anatidae. It employed metagenomic analysis techniques to assess bacterial diversity, identify antibiotic resistance genes, and evaluate biofilm formation by Escherichia coli isolates. The findings suggest differences in microbial composition between the bird species, with implications for conservation efforts and understanding the transmission of antibiotic resistance. Overall, the study provides valuable insights into the gut microbiota of migratory birds and their potential role in antibiotic resistance dissemination.
Introduction
l Line 61: “are” should be removed.
l I suggest to clearly state the main goals of the study in the introduction outlining briefly the key objectives of the research.
Results
l Line 100: “analysis” should be removed.
l Line 109: “Ligilactobacillus” and “Psychrobacter” should be in italics.
l Line 121: “Metastat” instead of “Matastat”.
l Line 164: “L. relictus” and “Anatidae” should be in italics.
l Line 181: I suggest the sentence should be revised to accurately reflect that the strains of E. coli were isolated from the fecal samples.
l Line 181: Regarding the number of the strains “117”, the sum of the number or strains in Table S1 is 116 and not 117. Please specify.
l The quality of some figures should be improved, particularly the clarity of small text. Some labels appear blurry and difficult to read, which may hinder understanding.
Discussion
l I recommend including a brief section discussing the limitations of the study and outlining potential future research directions.
Materials and methods
l Line 312: “effect” instead of “efect”.
l Line 331: Please provide the supplier’s information regarding sterile saline solution.
l Line 332: Please provide the supplier’s information regarding glycerol.
l Line 346: Please provide the supplier’s information regarding the enrichment solution liquid medium.
l Line 347: Please provide the supplier’s information regarding Macconkey agar.
l Line 380: Please provide the supplier’s information regarding the Luria-Bertani medium.
l Line 382: Please provide the supplier’s information regarding PBS.
l It would be helpful to provide a brief description of the sampling sites within the Ordos Honghaizi Wetland to provide a better understanding of the environmental conditions and potential sources of variation in the collected samples.
l Including information about the timing and frequency of sample collection (e.g., seasonal variations, sampling intervals) should add context to the study and help interpret the results. In the Discussion Section, for example, it is mentioned that samples were collected during the spring and fall seasons. I recommend that this should also be included in the Materials and Methods Section.
Comments on the Quality of English LanguageThe text should be checked from a native English speaker for grammatical and syntax errors.
Reviewer 2 Report
Comments and Suggestions for Authors
This study presents a compelling investigation wherein the authors employed metagenomics sequencing techniques to analyze fecal samples obtained from both L. relictus and Anatidae. Their analysis unveiled variations in the composition of intestinal flora across these avian species. Nevertheless, Firmicutes, Proteobacteria, and Actinobacteriota emerged as the predominant taxa in both cases. Additionally, according to data extracted from the Comprehensive Antibiotic Resistance Database, the authors illustrated that among migratory birds, glycopeptide resistance genes dominated in prevalence, accompanied by quinolone, tetracycline, and lincosamide resistance genes. Moreover, there was a notable decline observed in the abundance of resistance genes carried by migratory birds over the study period.
The scope of this paper is in line with those of Microorganisms and the analyses performed are scientifically sound. The paper is generally well written and structured.
However, I would recommend that the authors carefully revise the points/suggestions below:
1. Lines 103-126: I recommend merging the two Sections 2. 2 and 2.3.
2. Line 121: “Matastat analysis results” – should be “MetaStat analysis results”.
3. Lines 134-136: I recommend rephrasing the sentences as follows: “Among these samples, 76 resistance genes were consistent with those detected in the environment, with glycopeptide resistance genes being the most prevalent”.
4. Lines 150-151: Rephrase this sentence: “to visually observe the abundance ratio of ARGs in all samples, a selection was made to draw an overview circle featuring the top 10 ARGs”, considering the use of "depict" instead of "observe" and substituting "Chord diagram" for "overview circle".
5. Lines 170-172: “According to the findings of mobile genetic element prediction (Fig. 6), the ‘Isfinder’ category had the largest abundance and number of microbial groups in L. relictus and Anatidae faecal samples (Fig. 6A), while there were almost none in environmental samples”. Could you kindly consider rephrasing the following sentences in the manuscript text as they appear to lack clarity. It is crucial to note that according to the ISfinder database, the insertions sequences (IS) exhibited greater abundance in fecal samples from L. relictus and Anatidae compared to environmental samples.
Also, the ISFinder database needs to be mentioned in the “Materials and Methods” section as well in the text of the manuscript including the reference.
6. Lines 173-174: “Whereas other mobile genetic elements were abundant in L. relictus, Anatidae faecal and environmental samples were mainly ‘integrall’ (Fig. 6B) and ‘plasmid’ types (Fig. 6C)”. I recommend rephrasing the following sentences within the text for more clarity: According to the INTEGRALL database (INTEGRALL: a database and search engine for integrons, integrases, and gene cassettes), our analysis detected [insert names of detected items, e.g., integrons, integrases, gene cassettes, etc.] predominantly in environmental samples. Similarly, for plasmids, we urge the authors to specify the database used for plasmid detection and note that [insert names of detected plasmids if known] plasmids were primarily found in environmental samples.
Furthermore, it is essential to include references to the INTEGRALL and Plasmid databases in the 'Materials and Methods' section of the manuscript, as well as within the manuscript text. This will enhance transparency and enable readers to access the relevant resources.
7.Lines 176-179: In the legend of Figure 6(A): revise the title as follow: "Relative abundance of the Top 20 Insertion Sequences (IS) among samples".
Figure 6(B): revise the titles as follow: “Relative abundance of the top 20 integrases (if this is the case) [insert names of detected items, e.g., integrons, integrases, gene cassettes, etc.] across samples”.
Figure 6(C) revise as follow: “Relative abundance of the top 20 plasmids among samples”.
8. Lines 181-183: The sentences are not clear, rephrase as follow: “A total of 561 migratory bird fecal samples collected between 2021 and 2023 yielded 117 strains of E. coli. These isolates were characterized using a fully automated drug sensitivity system to determine drug resistance phenotypes. The resulting resistance patterns are presented in Table S1”.
9. Lines 189-190: Rephrase as follow: “In this study, E. coli isolates were categorised into four groups according to the biofilm-forming strength”.
10. Lines 194-203: I suggest a rephrasing of the section titled "Bacterial community functional prediction". For instance, when the authors stated, "The main function of the bacteria were cellular processes, environmental information processing, human diseases, metabolism, and organismal systems", it could be rephrased as follows:
"Functional prediction revealed that the bacterial community exhibited a predominant abundance of genes associated with cellular processes, environmental information processing, human diseases, metabolism, and organismal systems."
Alternatively:
"In the functional prediction of the bacterial community, genes related to cellular processes, environmental information processing, human diseases, metabolism, and organismal systems demonstrated a significant relative abundance."
More suggestions:
“Moreover, within the functional prediction of the bacterial community, genes associated with xxxx, xxxx, xxxx, and xxxx [insert the functions] were notably abundant”.
“The distribution of KEGG subclasses (level 1) indicated a higher occurrence of genes associated with various xxxx [insert the functions], followed by xxx [insert the functions]”.
11. Line 325: “abundance distribution circle maps” rephrase taking in consideration to use “Chord diagram” or “ring” instead of “circle maps”.
12. Figures:
- Figure 4 (A and B) and Figure 5 (C, D, E and F): I suggest enhancing the contrast and visibility of the figures in the manuscript text to facilitate reader comprehension. Please check the visibility of text annotation, key, title of axis etc.
- Figure 2: In axis title of Figure 2C, correct: “relative abundance at the genus level”. In the Figure caption add the following: “ES: environmental samples”.
- Figure 5:
- In the legend Figure (E), the sentence “Overview circles plotted by antibiotic resistance ontologies with maximum abundance of the top 30 resistance genes” appears to be correct, but it could benefit from a slight adjustment for clarity, I would suggest “Plotting Chord diagram illustrating the antibiotic resistance ontologies with maximum abundance of the top 30 resistance genes”. Authors may use “ring” instead of circle in this type of diagram.
- In the legend of Figure (F): the sentence “Distribution map of resistance mechanisms based on the mechanism of action of resistance genes in relation to species” appears to be correct, but it could benefit from a slight adjustment for clarity: “Chord diagram showing the distribution of resistance mechanisms based on the mechanism of action of resistance genes relative to species”.
In these two Chord diagrams (Figure 5 E and F), I suggest that the authors provide additional details regarding the representation of the length of the bars on the outer ring, the color of the ribbons, and the width of each ribbon.
- Figure 6: In Figure 6 (A) and (B), the authors presented GenBank numbers (Figure 6A) and NCBI Reference Sequences (Figure 6B) as part of their investigation utilizing the INTEGRALL database or the plasmid database (though the specific name of the database is not provided in the text or Materials and Methods section). I recommend including the names of the integrases and plasmids in the Figures, if applicable, to enhance clarity. Additionally, the reference for this information could be provided elsewhere in the manuscript.
13. Tables:
- Table S1: revise the title as follows: “Statistics of antibiotic susceptibility testing of Escherichia coli”.
Comments on the Quality of English LanguageWhile I have already rephrased some sentences in the manuscript text (please refer to the report), I encourage you to further refine sentence structures to enhance clarity. Ensure that each sentence conveys its intended meaning with precision. Additionally, please pay careful attention to grammar and punctuation to eliminate any inconsistencies. It is also important to ensure smooth transitions between paragraphs for improved flow.
Round 2
Reviewer 1 Report
Comments and Suggestions for Authors
The manuscript has been significantly improved and I think it is now suitable for publication.
Comments on the Quality of English LanguageMinor editing of English language is required.